# Costs of Palliative Care in Oncological and Non-Oncological Patients with Different Types of Ambulatory-Based Attention: Cost-Study Protocol

**DOI:** 10.3390/diseases12100243

**Published:** 2024-10-05

**Authors:** Ana Helena Perea-Bello, Marta Trapero-Bertran, Christian Dürsteler

**Affiliations:** 1Faculty of Medicine and Health Sciences, Universitat de Barcelona, 08036 Barcelona, Spain; 2Department of Economics and Business, Faculty of Law, Economics and Tourism, Universitat de Lleida, 25001 Lleida, Spain; marta.trapero@udl.cat; 3Department of Anaesthesiology, Consorci Sanitari de l’Alt Penedès-Garraf Sant Pere de Ribes, 08810 Barcelona, Spain; dursteler@ub.edu; 4Department of Surgery and Surgical Specializations, Faculty of Medicine and Health Sciences, Universitat de Barcelona, 08036 Barcelona, Spain

**Keywords:** palliative care, costs, outpatient care, oncological, non-oncological

## Abstract

**Background:** Ambulatory-based palliative care is vital to managing oncological and non-oncological patients. Its economic impact on the healthcare and social system has recently begun to be considered significant. It is essential to agree on the cost types, the methodology for approaching and analyzing these costs, and how to determine the burden imposed by this attention on the healthcare and social system. **Aim:** This study aims to design a study on the economic burden of palliative care (PC) in oncological and non-oncological pathologies in the context of outpatient care (ambulatory-based and home support teams). **Methods:** A prospective cross-sectional study on the economic burden of ambulatory-based palliative care (ABPC) in three phases is conducted. **Phase I:** A systematic literature review (SLR) first defines the methodology and data to acquire for costing (results already published). **Phase II:** The next phase is the piloting of the registration questionnaires for costs/expenses (results already analyzed and presented). **Phase III:** A cross-sectional study is being conducted to collect data on the direct and indirect costs of ABPC assumed by the healthcare system and patients/caregivers to estimate its economic and social burden (in progress). **Discussion:** In this study, we create and propose a methodology and extend the approach to the funding of PC in an ambulatory-based context to determine its social cost and provide stakeholders with more information to assign resources more efficiently.

## 1. Introduction

With the global demographic change [1] due to the aging of the population, changing social and health needs must be addressed to facilitate quality care at the end of life [2].

Palliative care (PC) is an area of health practice and growing academic activity, serving individuals of all ages with advanced and progressive diseases—both oncological [3] and non-oncological [4]—with the aim of improving their quality of life and end of life.

The World Health Organization (WHO) states that palliative care “is an approach that improves the quality of life of patients (adults and children) and their families who are facing the problems associated with life-threatening illness, through the prevention and relief of suffering by means of early identification and correct assessment and treatment of pain and other problems, whether physical, psychosocial or spiritual…” [1], and the European Association for Palliative Care (EAPC) adds that they “…affirm life and consider death as a natural process. They are managed to maintain the best possible quality of life until death” [5].

The costs of curative interventions in oncology, internal medicine, and areas such as neurology or geriatrics—amongst others where curative treatments for specific conditions exist—are well documented, and every model of the healthcare system has a specific methodology for registering and billing the expenses [6]. However, when PC is indicated, the costing and billing of the interventions, including those carried out during hospital stays as well as on an outpatient basis or at home, are not collaboratively arranged and are usually addressed from the healthcare system perspective—public or private. From this perspective, most of the indirect and intangible costs assumed directly by the patients and their caregivers are not considered, resulting in an important data gap when making final calculations. Hence, essential interventions intended to improve quality of life during the PC period—medications, physiotherapy, psychotherapy, social work advice, outpatient visits, and formal caregivers—are not properly recorded and invoiced, which leads to the incomplete calculation of PC service costs and the undervaluation of the costs of PC actions and practices with consequent underpayment [7,8].

Studies on the burden of disease seek to identify, measure, and evaluate the costs of health and non-health resources and productivity losses from a certain perspective and in a specific period. To do this, it is necessary to identify, measure, and value the resources used to prevent and treat disease, using available sources that best reflect the opportunity cost of said resources, as well as other resources whose usual use has been modified for this purpose. This information can be useful as a basis for economic evaluation or to help decision-makers prioritize health policies and interventions when considering efficiency. Currently, most studies on the burden of disease include formal healthcare costs; however, there is growing interest in incorporating the burden of informal care into unrelated costs [8,9].

The institution of formal PC—a health service delivered by trained personnel—is a relatively recent health service offered to the population [1,10]. As such, its resource consumption has been studied less than the usual care scenario—stopping curative interventions when they are no longer indicated—so quality evidence supporting PC costing is scarce in the literature [7,11,12]. In fact, the medical literature reports that PC interventions reduce the total costs of health interventions; however, most published studies are not comparative, prospective, or clear enough regarding the comparison they are making (i.e., if a service model or a billing model is being compared).

From a health system perspective, PC costs are generally measured by reviewing the direct administrative costs recorded in the hospital or health center because they are traditionally the most easily accessible (Diagnosis-related Group [DRG] system) [13]. However, the indirect costs—from informal PC provided by family/caregivers—are seldom recorded during studies, but when they are, they represent a large proportion of the costs during the last months (even weeks) of life—between 27% and 34% of the total expenditure in that period [12,14].

The economic impact of a PC team managing a chronic or terminal condition with the support of the patient’s family/caregivers has not been studied in depth, considering that the typically measured costs do not include informal care (i.e., social support, night shifts, transportation, appointment planning, medication preparing and giving, pharmacy visits, etc.). Furthermore, cost studies are framed in specific contexts and are limited by the availability of data, which limits their generalizability, although not their importance [14].

In conclusion, to analyze the economic burden of PC in oncological and non-oncological conditions, according to the type of care offered in our context—ambulatory in outpatient clinics or home support teams coordinated by a reference center or hospice—we must obtain data on direct health and non-health costs to obtain a broader picture and determine the burden of care imposed on the health system, the patients, and their families when delivering this PC modality.

We aimed to design a study on the economic burden of PC in oncological and non-oncological pathologies in the context of outpatient care (ambulatory-based and home support teams) and propose a structured methodology for researching this topic.

## 2. Aims

### 2.1. General Aim

This research aims to design a study on the economic burden of PC in oncological and non-oncological pathologies in the context of outpatient care (ambulatory-based and home support teams).

### 2.2. Specific Aims

The following are this study’s specific aims:-Identify the relevant costs to include in a PC cost/economic burden study.-Identify different methodologies for calculating burden/economic costs in PC.-Determine and compare the costs of PC in oncological and non-oncological pathologies.-Propose a methodology to identify the relevant costs, how to analyze them, and the most convenient perspective from which to study the burden of disease in outpatient PC.

## 3. Study Hypotheses

**H1** **:**
*The costs of formal and informal palliative care in the most common oncological and non-oncological conditions in Spain (Section 4.2.1) comprise at least 40% of healthcare costs in the last year of life (of the patients under study) [15,16].*


**H2** **:**
*The costs of formal and informal palliative care in oncological pathology are lower than those in non-oncological pathology.*


## 4. Methods and Analysis/Study Design/Setting and Recruitment

### 4.1. Study Design

We proposed a multicenter, national (Spain), prospective study on the economic burden of palliative care in the outpatient setting in an outpatient clinic and home care team in a third-level hospital and in a hospice. It is being carried out in three phases:

#### 4.1.1. Phase I: Systematic Literature Review (SLR)

An SLR [17,18] was performed to achieve the following:-Identify the costs and methodologies proposed to identify and measure the burden of PC.-Design questionnaires to record costs assumed by the health system (data from the clinical record [DRG]) as well as by patients and their relatives/caregivers (data from questionnaires and interviews with patients and their relatives/caregivers) [19].

#### 4.1.2. Phase II: Pilot Study

Phase II includes piloting patient/caregiver recruitment, interview technique, and cost registration questionnaires (corrections thereof) and revising the billing methodology of participating institutions (DRGs).

#### 4.1.3. Phase III: Cross-Sectional Cost Study (In Progress)

Phase III includes the following:-Cost data collection (from institutional billing [DRG] and from patients/caregivers).-Analysis of the data collected on costs and calculation of the economic burden of PC (the analysis depends on the data collected).-Proposal for a methodology to study costs in the context of PC.

### 4.2. Sample Size

#### 4.2.1. Population

The population being studied includes patients of legal age, without an age limit, who are affected by breast, colorectal, or lung cancer; chronic heart failure (CHF); or chronic obstructive pulmonary disease (COPD), and their families/caregivers who are receiving PC on an outpatient basis (in Spain).

#### 4.2.2. Sample Size Calculation

-For *Phase II*, we planned to recruit, by convenience sampling, 3–5 patient/caregiver pairs with the diagnoses of interest who meet the inclusion and exclusion criteria and agree to participate.-For *Phase III*, patient/caregiver pairs will be also recruited by convenience sampling from those who meet the inclusion criteria. The number of patient/caregiver pairs to recruit in Phase III depends on the following parameters:

For *expenses assumed by the health system*

**Step 1.** According to the distribution of cases of breast, colorectal, and lung cancer presented by Global Cancer Observatory - Globocan in 2020 and 2024 [20], each contributes between 10 and 13% of the total incidents. In the PRICE study, a CHF prevalence of around 7% is reported in people over 45 years of age [21], and data from the Global Burden of Disease (GBD) show that COPD is the third-most common cause of mortality [22]. Considering the above, plus a 10% adjustment for potential losses, we will need 12–13 patient/caregiver pairs with each type of cancer, CHF, or COPD.

Once the planned number of units (patient/caregiver pairs) is achieved, we must check whether this number is sufficient for the mathematical model (**Step 2**).

**Step 2.** Data from the study by Guo et al. in 2018 [8] are collected to build a prediction model for CP expenses, and for this objective, a minimum of 50 + (8 * m) cases are required for the test hypothesis (multiple correlation is zero, power 80%, with alpha 5% and a medium effect in the regression analysis (R2 = 0.13), where m is the number of predictors). The predictor variables will be defined during the SLR.

For *expenses assumed by the patient or family/caregivers*

Gott et al. in 2015 [23] and Gardiner et al. in 2016 [9] published data on indirect costs during patient care in PC and collected the data from a group of 30 relatives of patients who recently died from cancer or other non-oncological pathologies [24,25,26]. This number will be considered the minimum necessary.

## 5. Data Collection Strategy

### 5.1. Phase I. Systematic Literature Review (SLR) [27] (Figure 1)

From the SLR, we concluded that the number of studies on the costs of ambulatory PC is increasing. Most were conducted from a healthcare system perspective, which omits costs related to patients’/caregivers’ economic burden. Based on the evidence found, we verified that productivity losses for PC patients and their caregivers are the least-studied cost impact, so future research should attempt to measure and economically quantify such losses. 

With data on the types of costs registered during the revised studies (Table 1), we constructed a questionnaire to collect detailed data on patients’ and caregivers’ health and the costs associated with ambulatory-based PC (Appendix A). We propose that future studies on this topic be prospective and make cost calculation approaches using societal and patient perspectives to calculate the financial burden more precisely and evaluate interventions designed to improve the quality of life of patients under PC [27].

Once the SLR was published, we proceeded with the piloting of the cross-sectional study.

### 5.2. Phase II: Pilot Study

The aims of this phase were to test the following: i. the methodology for recruiting patient/caregiver pairs; ii. the adequacy and completeness of the questionnaire in collecting data on the costs of PC; iii. the acceptability of the interview session (face-to-face or by telephone) by the patients and caregivers; and iv. the ease of collecting and filing the data on the REDCap^®^ platform.

The pairs of participants were invited by physicians from the participants’ institutions and signed informed consent [IC] forms with them. Before the invitation was given, it was confirmed that both patients and caregivers were in suitable physical and emotional conditions to participate.

The interview/questionnaire involved a maximum of two (2) sessions—in person or by telephone—that were different from those scheduled with the referring physician (oncologist, cardiologist, family medicine, pulmonologist, or palliative care doctor). Interviews for data collection were conducted by the principal investigator (*AHPB*).

In the cases of patients who died after giving IC but before the interview and who had family members who agreed to participate, their willingness to continue in the study was explored [8].

We recruited four patient/caregiver pairs (Table 2). One patient died before the interview, and his caregiver preferred not to participate. Of the three remaining pairs, one patient allowed a face-to-face interview, and two accepted a telephone interview; we were able to contact only two caregivers. In the end, three patients (one female and two males) and two caregivers (both females) were interviewed.

Patients and caregivers accepted recruitment well. All reported a preference for a telephone interview because of its ease. One caregiver (male) was contacted three times by phone but did not allow the interview (i.e., “*not a good moment*”).

Patients found the questions easy to answer. They provided good quality data on demographics and health state but did not provide all the data on the costs of their care (e.g., “*My wife knows it better*,” “*My wife manages the money”*). On the caregivers’ side, all found the questions on the patient’s and their own health clear and straightforward to understand and answer. Data on costs were not easy to remember or calculate, as in the case of patients, but caregivers were more conscious of the expenses compared with patients.

All the interviews took less than an hour and were well tolerated. In one case, a second session was necessary because of the patient’s fatigue (male, COPD).

Registering data while interviewing was possible using REDCap^®^, and quick corrections on the questionnaire were allowed without losing data.

Patients’ and caregivers’ insights significantly influenced the data on indirect healthcare costs and patient expenditures. Their feedback led to corrections for the *Phase III* logistics [71].

### 5.3. Phase III: Cross-Sectional Study (In Progress)

This phase is the last part of the study, constructed after obtaining the results of *Phases I* and *II* and started in April 2024.

In summary, with the Ethical Research Committee’s approval, the SLR was completed and published, and the piloting results were revised and presented, so we proceeded with *Phase III* (cross-sectional study on costs).

The logistics in *Phase III* are the same as in *Phase II* once corrections in algorithms and questionnaires are made.

*Phase III,* then, has two parts—*recruitment* and *interviewing*. The featured *recruitment* and *interviewing* routes have been revised, are being followed, and will have minor modifications depending on their acceptability by the participants. The inclusion of patients is a collaborative effort, and the route can be adjusted by the participants’ institutions, clinical colleagues (Hospital de Granollers, Hospital Clínic de Barcelona), and institutional home care teams (PADES), depending on their workplace dynamics (Figure 2).

## 6. Data Collection Questionnaires

For *Phase II* (piloting) and *Phase III* (cross-sectional study), the data collected during the interview with each patient/caregiver are stored using REDCap^®^ through Hospital Clínic’s platform (https://redcap.clinic.cat/). As decided after the SLR, the data to be collected include the following (Table 1):-Direct healthcare system costs (DRG, adjusted for every institution).-Direct non-healthcare system costs.-Direct healthcare costs for the patient.-Productivity losses of the patient.-Productivity losses of the family/caregiver.-Other costs (e.g., out-of-pocket expenditures).

All data are collected using the REDCap^®^ questionnaire (Appendix A) and are stored in the REDCap^®^ platform.

## 7. Data Analysis

### 7.1. Phase I: Systematic Review of Literature on Costs

The PRISMA methodology was followed. A methodical and directed search of biomedical literature was conducted in pre-selected academic databases. The articles that met the selection criteria were obtained, and a critical reading was conducted [72]. The relevance and quality of the chosen articles were determined using the QHES instrument [27,73].

### 7.2. Phase II/III

-
*For the data on expenses assumed by the health system, the following is planned:*


The data collected by the health system (DRG) and the methodology used to code and subsequently charge for the services provided to patients will be reviewed [74].

-
*For data on expenses assumed by family/caregivers, the following is planned:*


The data collected through the interviews will be reviewed to determine the category of costs to which they should be assigned for analysis.

For data collection and analysis, the terms/variables that prove useful in the definition of costs and their calculation, which are recorded through the literature review, billing review, and interviews, will be categorized and used to analyze the collected data (data will be analyzed and modeled to report the economic burden of the PC service).

We also plan to explore the acceptability of the interview/questionnaire session. The acceptability will relate to the medium (face-to-face vs. telephone interview), time (<1 h vs. >= 1 h), and ease of comprehending and answering the questionnaire (not intended to be a qualitative study or a study of the operative characteristics of a diagnostic test).

The data will be reviewed after *Phase II* (piloting), and the following will be determined:-Whether the *demographic/clinical* data are sufficient and have been collected adequately. Measures of central tendency and dispersion will be determined for numerical variables; proportions and differences in proportions will be determined for categorical variables. A description of the data collected will be made (inferences cannot be made with the pilot sample).-Whether the record of *billing data* from the *medical history* and the *expenses incurred by the patient and caregiver* are complete and clear. A descriptive analysis of these data will also be carried out.

Once the data have been reviewed, errors will be corrected, and the questionnaires will be adjusted. If there are doubts about the completeness of the data and their relevance, the related literature will be reviewed, and new adjustments will be made before proceeding with *Phase III* (cross-sectional study).

## 8. Discussion

The novel aspect of this study is the proposal to build the methodology to acquire cost data on PC from zero. First, we performed a systematic review to determine the types of costs, cost outcomes, and costs included. We explored the methodology used by the authors to procure the data from the healthcare system, patients, and caregivers.

We plan to collect and analyze data separately for oncologic and non-oncologic patients, hypothesizing that costs differ based on the condition’s nature.

Also, we decided to approach patients and caregivers to gain more in-depth data after detecting that caregivers are most often the planners of ABPC. Along the same line, we maintain the aim/proposal of measuring, tagging, and registering the informal caregiver’s time spent providing care in detail.

This planned approach is different from most of the other detected studies that address the topic (SLR). These studies have no specific research methodology and use a third-party perspective that does not consider most of the indirect costs and the role and burden assumed by the patients and caregivers in the delivery of ABPC.

With this study, we sought to develop and propose a methodology and extend the approach to PC funding in the outpatient context. We also aimed to make the social cost of outpatient PC evident and give stakeholders more information to assign resources more efficiently.

## 9. Conclusions

By presenting this protocol as a manuscript, we can report that cost studies on ambulatory PCs are increasing. They are mainly conducted from the healthcare system perspective and have a considerable lack of reported data on costs related to patients’ and caregivers’ economic burden.

Having completed two phases, we are convinced that acquiring cost data from the healthcare system, patients, and caregivers is the key to producing robust data. This data will not only inform stakeholders who allocate monetary resources but also has the potential to revolutionize the management of ambulatory-based PC, making it more efficient and socially beneficial.

## Figures and Tables

**Figure 1 diseases-12-00243-f001:**
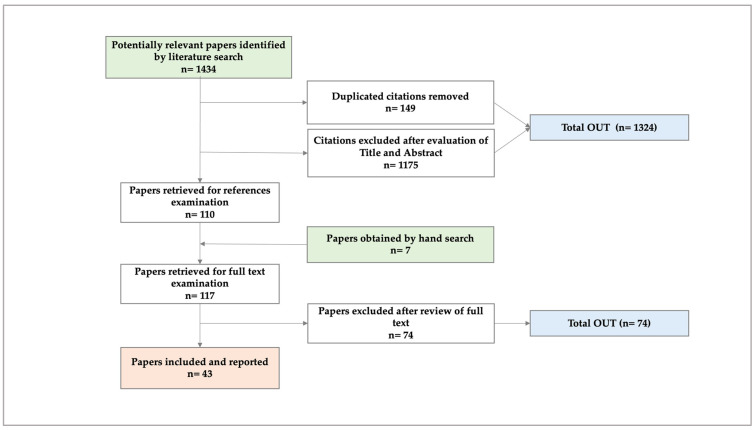
Preferred Reporting Items for Systematic Reviews and Meta-Analyses (PRISMA) flowchart of the SLR.

**Figure 2 diseases-12-00243-f002:**
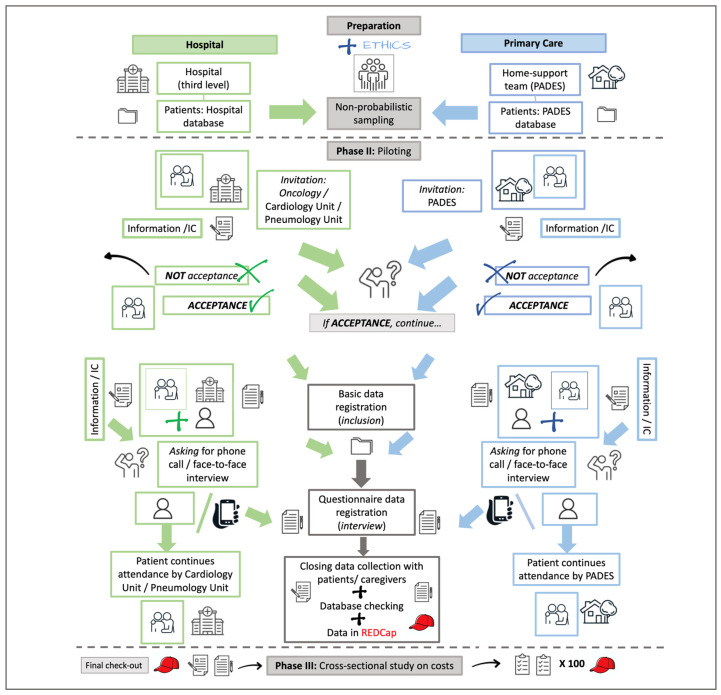
Flowchart of the intended route to *recruit* and *interview* patients and caregivers from hospital attention and home care teams. PADES: *Programa de Atención Domiciliaria y Equipos de Soporte*; IC: Informed Consent (part of the images used were taken from *iStock* [free images]; https://www.istockphoto.com/es).

**Table 1 diseases-12-00243-t001:** Cost data to collect through interviews with patients/caregivers ^a^ (modified from [27]).

Cost Type	Components ^b^	References
* Direct healthcare system costs *	Primary physician visit	[12,14,28,29,30,31,32,33,34,35,36,37,38,39,40,41,42,43,44,45,46,47,48,49,50]
	Specialist physician visit	[12,28,29,30,32,33,34,35,36,37,38,39,40,42,43,44,45,49,50,51,52,53,54,55,56,57,58,59,60,61,62,63]
	Specialist nurse visit	[12,28,29,30,32,34,37,41,42,43,51,53,54,56,57,62,64,65]
	Physical or occupational therapy visit	[12,28,33,34,38,41,42,43,44,54,56,58,62,64]
	Emergency visits	[12,28,30,34,36,37,39,40,41,43,46,47,51,54,56,58,60,63,66,67,68]
	Day-care hospital	[30,39,41,43,50,53,60,63,69]
	Domiciliary attention	[29,30,31,32,34,35,36,37,38,39,40,41,42,43,44,47,49,50,51,52,54,55,56,60,63,66,69]
	Medication	[14,28,29,31,32,33,34,35,36,37,38,40,44,52,55,57,58,60,64,68,69]
	Ambulatory procedures	[58,60]
	Equipment	[12,32,34,35,36,37,38,42,44,50,55,58,64]
	Diagnostic tests and images	[33,34,36,39,57,69,70]
	Acute inpatient attention	[12,14,29,30,31,32,33,34,35,36,37,38,39,40,42,44,45,46,47,49,50,51,52,53,55,56,57,58,60,61,62,63,64,66,68,70]
	Medicalized transportation	[14,32,33,35,38,53,63,64,66,70]
* Direct non-healthcare system costs *	Professional care out of home (hospice, nursing home, day center)	[43,63]
	Professional care in home (12/24 h nurse or professional caregiver)	[36,43,47,55,62,63]
	Various sanitary services (occupational therapy, psychology, social work, nutrition)	[43,62,63]
	Other informal care (spiritual)	[12,32,34,35,36,38,43,57]
	Home adaptations	[43]
* Direct healthcare cost (patient) *	Co-payments	[12,28,32,33,35,36,38,57]
	Private healthcare services (appointments)	[28,34,36]
	Other healthcare professionals (private practice)	[28]
	Medications/equipment not covered or partially covered by public/private service	[12,32,34,36,55,57]
	Private transportation	[28,33,34,36,55,57]
	Home adaptations paid by patient/caregiver	[12,28]
* Productivity loss (patient) *	Work absenteeism	[14,28,70]
	Temporary occupational leave	[14]
	Permanent occupational leave	[14]
	Early retirement	-
	Work decline	-
	Total or partial loss of income (self-employed workers)	-
*Productivity loss (family/caregiver)*	Work absenteeism	[14,28]
	Temporary occupational leave	[14,34,35,36,38,55,57]
	Permanent occupational leave	[14]
	Time devoted to care	[14]
	Loss of leisure time	[12,14,34,36,55,57]
*Others*	Out-of-pocket expenditures	[12,28,32,34,35,36,38,43,55,57,70]

^a^ The data will be obtained through face-to-face or telephone interviews and filed using a REDCap^®^ questionnaire. ^b^ Each one of the components to explore was included after RSL.

**Table 2 diseases-12-00243-t002:** Data of pilot study (*Phase II*).

	Patients (n = 4)	Caregivers (n = 4)
**Gender**		
Female	1	3
Male	3	1
**Condition that motivates ABPC**		
Oncologic	2	N/A
Non-oncologic	2	N/A
**Recruitment**		
By main researcher	3	2
By secondary researcher	1	2
**Recruitment via**		
Face-to-face	4	2
Telephone	1	2
**Questionnaire appliance via**		
Face-to-face	1	-
Telephone	2	2
**Completion of questionnaire (yes)**	3	2
**Data on costs of healthcare system (yes)**	3	2
**Data on costs for patients/caregivers**	3	2

## Data Availability

Data availability is not applicable in this phase, except for those related to the already published SLR (***Phase I***) [27]. At the end of the study (end and publication of results of ***Phase III***), data will be available from the corresponding author on request.

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
