# Peer review of "Costs of Palliative Care in Oncological and Non-Oncological Patients with Different Types of Ambulatory-Based Attention: Cost-Study Protocol"

_diseases, 2024, doi:10.3390/diseases12100243_

Round 1

Reviewer 1 Report

Comments and Suggestions for Authors

Thank you for the opportunity to review this protocol paper on a very important topic – costs of palliative care for patients and carers.  This study potentially fills an significant gap in the literature and in health care policy and financial considerations.

I note that there are some minor grammar and punctuation errors that may be due to Spanish to English translation.

There are main things that I suggest to improve the clarity of paper.

1.      In the abstract make it clear that the first two phases have already been undertaken and the third is underpay. That will make the change of tense more understandable.

2.       I suggest changes to figure 2a to make it clearer that this is about recruitment

3.      I am unclear exactly what data is being collected and analysed in RED Cap, and how, in the third phase. Some additional explanation would help explain Figure 2b in more detail.

Minor typos / English language grammar changes, for example Page 2 line   -‘de’ – change to ‘the’ and punctuation.

Page 2 line 80 – which population? The international population or Spain in particular?

Page 3 study hypothesis – I am unsure where the 40% of the costs comes from.

Page 4 line 153 – conveniently may be deleted for Phase II. Is it 3-5 patients/caregivers for each of the diagnoses of interest, or 3-5 patients and caregivers in total.
 page 4 – for expenses assumed by the health system -  does this heading fit under Phase III? Can the authors explain why two options are provided here. Is it that either may be used depending on Phase I and II, or both will be used?

Page 5

 Is the protocol paper being published after the research has been undertaken? In the abstract it reads as if the study is to be undertaken ‘This study seeks to…suggest future tense) , but the systematic literature review has already been completed and published, and Phase II pilot study  is in past tense. I suggest identifying in the abstract that the first two phases are complete and the third is underway.  

Page 5 – I am unclear who the regular doctor is and why they were invited to attend  to provide additional support. Was it the participants desire, or the doctor’s desire for them to attend?

Page 5 – what is the background /clinical role of the principal investigator who was conducting the interviews?

Page 6, Table 1

 Suggest left hand alignment of first column rather than centred for easier reading.

What does (yes) signify?

Perhaps identify in the table that one patient died, so that the numbers add up.

Page 6

 I am unsure what the chosen route refers to? Do you mean data collection option (may be a Spanish-English translation consideration). Are you referring to the face to face or telephone interviews?

216 – is the SLR not already  completed and published as per Figure 1?

Page 7  Figure 2a  Flowchart. Do the authors mean attendance rather than attention?

In Figure 2a – is the preparation phase about recruitment? And is piloting still about recruitment or data collection? Perhaps put recruitment in the Figure 2a title to make this clearer.

 In Figure 2b, I am unsure what secondary complementary data registration means.

Also does green still mean hospital and blue primary care? Perhaps put those headings in this figure also.

Page 8 – is billing data only collected from patients and carers, or also from health care billing records? Some more detailed information here would be useful. What data exactly is going into RedCap?

References: Relevant and correct.

Overall, this looks like a very worthwhile study.

Comments on the Quality of English Language

I note that there are some minor grammar and punctuation errors that may be due to Spanish to English translation.

Author Response

Respectful regards,

In the attached PDF file is the review of the comments, with our respective responses.

The response should be revised with the corrected version of the manuscript.

Ana H. Perea B.

Reviewer 2 Report

Comments and Suggestions for Authors

The authors are planning to conduct a three-stage study addressing economic burden of palliative care (PC). For this aim here they present their study protocolI. my overall opinion is that it is an interesting topic. For example, I have appreciated the inclusion of both direct and indirect costs, and the consideration of both formal and informal PC. Consequently, I have only few suggestions that could improve the manuscript quality.

-The aim of the planned investigation should be clarirly stated at the beginning of the abstract. Therefore, the reader is immediately informed of the study's focus.

-While the Introduction section offers a solid bakground, in my opinion, it should be shortened. Additionally, a more organized narrative flow is requiredow. The authors should start with the importance of PC, followed by gaps in current research on economic burders, and concluding with the study’s aim.

-Conclude the Introduction with the research query.

-I suggest to revise the language style for improving readibility. Specifically, the sentence are too long and complex.

-Does Figure 1 refer to a previous EBM study from the authors? If yes, the flowchart should be removed and yhe authros can report the results in the text.

- Methods: some parts should more concise. For example, the detailed explanation of each phase could be streamlined to underline the most critical aspects.

-The pilot study’s results should be better discussed. It could provide context for the main study.

-Lines 196,197: add references.

Comments on the Quality of English Language

Revise as suggested

Author Response

(The authors gave the same response as above.)
